# Mapping structure heterogeneities and visualizing moisture degradation of perovskite films with nano-focus WAXS

Nian Li [1,7], Shambhavi Pratap[1,7], Volker Körstgens [1], Sundeep Vema [2,3], Lin Song[4], Suzhe Liang [1], Anton Davydok[5], Christina Krywka[5] & Peter Müller-Buschbaum [1,6] ✉

Extensive attention has focused on the structure optimization of perovskites, whereas rare research has mapped the structure heterogeneity within mixed hybrid perovskite films. Overlooked aspects include material and structure variations as a function of depth. These depth-dependent local structure heterogeneities dictate their long-term stabilities and efficiencies. Here, we use a nano-focused wide-angle X-ray scattering method for the mapping of film heterogeneities over several micrometers across lateral and vertical directions. The relative variations of characteristic perovskite peak positions show that the top film region bears the tensile strain. Through a texture orientation map of the perovskite (100) peak, we find that the perovskite grains deposited by sequential spray-coating grow along the vertical direction. Moreover, we investigate the moisture-induced degradation products in the perovskite film, and the underlying mechanism for its structure-dependent degradation. The moisture degradation along the lateral direction primarily initiates at the perovskite-air interface and grain boundaries. The tensile strain on the top surface has a profound influence on the moisture degradation.

Hybrid perovskite materials have demonstrated extraordinary progress in optoelectronics such as photovoltaics[1], light emission diodes[2] and photodetectors[3,4]. The great success originates mainly from their broadly tunable chemical compositions (e.g., $APbX_3$, A = $CH_3NH_3^+$ ($MA^+$) or $CH(NH_2)_2^+$ ($FA^+$); X = $Cl^-$, $Br^-$, or $I^-$)[5]. Through compositional engineering of perovskite precursors, one can efficiently adjust the physio-chemical properties of the resultant perovskites, giving rise to high-performance devices. For example, $(MAPbBr_3)_x(FAPbI_3)_{1-x}$ perovskite films reached high solar cell efficiencies >20%[6,7] (the best perovskite solar cell to date has a power conversion efficiency of 25.7%[8]) rendering perovskite solar cells even attractive for space applications[9]. Although mixed hybrid perovskites display exceptional optoelectronic properties, they potentially suffer from residual strains. Especially, the tensile strain is a source of instability[10,11] and can accelerate the degradation of perovskite materials[12]. Emergent residual strains not only originate from external conditions such as thermal stress during the crystal growth, but are also closely relevant to local lattice mismatches/distortions, grain boundaries or local lattice misorientations[12–14]. In addition to causing intrinsic stability issues, recent studies demonstrated that the residual strain is detrimental to charge carrier dynamics in perovskites[15,16].

[1]Lehrstuhl für Funktionelle Materialien, Physik-Department, Technische Universität München, 85748 Garching, Germany. [2]Department of Chemical Engineering, Indian Institute of Technology Delhi, Hauz Khas, New Delhi 110016, India. [3]Department of Chemistry, University of Cambridge, Lensfield Road, Cambridge CB2 1EW, UK. [4]Frontiers Science Center for Flexible Electronics (FSCFE) and Xi'an Institute of Flexible Electronics (IFE), Northwestern Polytechnical University (NPU), Youyixilu 127, Xi'an 710072 Shaanxi, China. [5]Helmholtz-Zentrum Hereon, Max-Planck-Straße 1, D-21502 Geesthacht, Germany. [6]Heinz Maier-Leibnitz-Zentrum, Technische Universität München, 85748 Garching, Germany. [7]These authors contributed equally: Nian Li, Shambhavi Pratap. ✉e-mail: muellerb@ph.tum.de

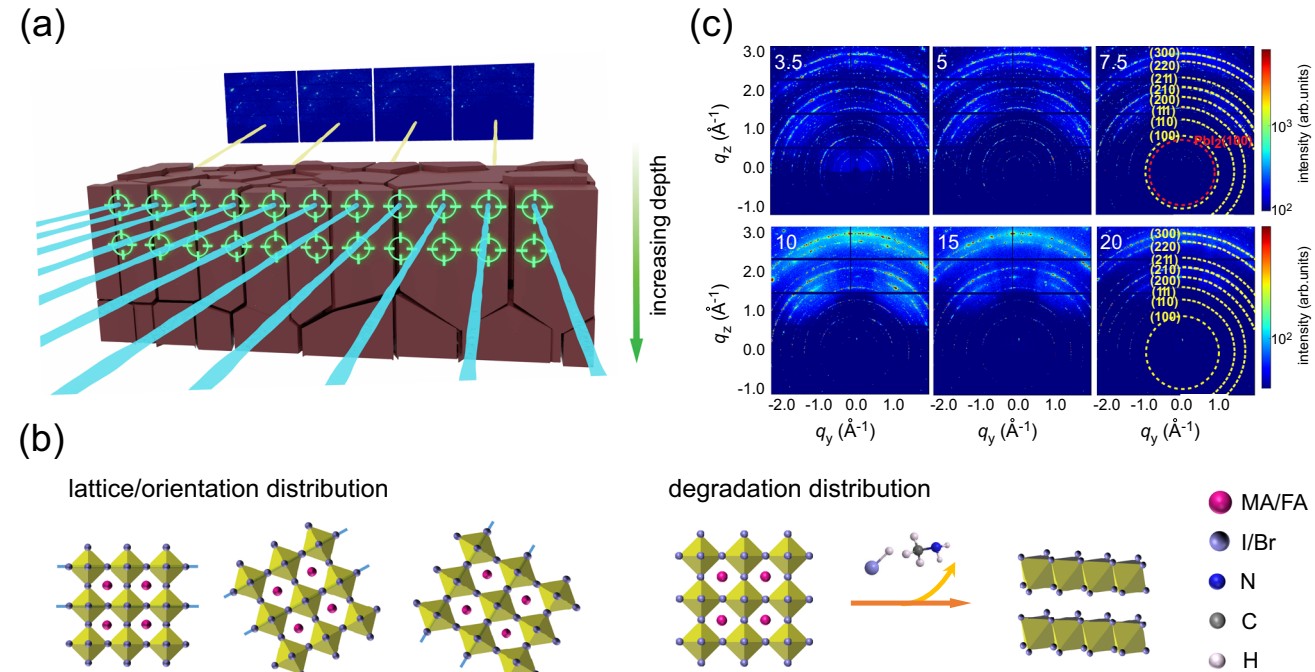

**Fig. 1 | Analysis of nWAXS mapping of the perovskite film. a** Schematic illustration of nWAXS scans of the $(MAPbBr_3)_{0.50}(FAPbI_3)_{0.50}$ film. The incoming X-ray beam is indicated with blue lines, and the outgoing X-ray beam is indicated with yellow lines. The scanning area is 80 μm length × 20 μm depth with a well-defined step-size of 500 nm. **b** Schematic highlighting the focus of nWAXS mapping of the $(MAPbBr_3)_{0.50}(FAPbI_3)_{0.50}$ film: lattice/orientation distribution and moisture-induced degradation distribution. **c** Selected depth-dependent 2D nWAXS patterns at 3.5, 5, 7.5, 10, 15, and 20 μm. Each image is summed over 160 frames, scanning a large length scale of 80 μm. The diffraction pattern rings, which correspond to a cubic perovskite phase, are indicated with yellow, and the diffraction ring representing the $PbI_2$ (100) peak is indicated with red.

The crystallographic orientation of perovskites is another factor that plays an important role on charge transport within the films[17,18]. Thus, an in-depth understanding of the atomic-scale structure of mixed perovskites is imperative for substantial improvements in stability and efficiency. In order to investigate such crystal information especially for the inter-planar spacing ($d$; corresponding to the scattering vector $q$) and crystallographic orientation, grazing-incidence wide-angle X-ray scattering (GIWAXS) and X-ray diffraction (XRD) are widely used characterization methods[11,15,19–21]. However, these conventional techniques determine an ensemble average of structural features probed over a macroscopic area[22]. Thus, the local crystalline structure heterogeneity of perovskite films cannot be analyzed with GIWAXS and XRD. Importantly, depth profiles along the surface normal of the film, providing a thickness dependent local information, to the best of our knowledge, have not been studied unambiguously.

Moreover, perovskite films are well-known to be sensitive to external stimuli like moisture, which induce the film degradation[11]. Initially, water ($H_2O$) incorporation into perovskite materials leads to the formation of monohydrates, which then degrade further to form dihydrates[23]. Upon long-time moisture exposure, irreversible degradation products, i.e., $PbI_2$, are formed[24]. So far, some works have shown that the permeation of moisture into perovskite films can easily take place at certain spots, such as the grain boundaries or crystallographic defects[24–26]. For instance, Wang et al.[24] and Yun et al.[27] used local real space imaging techniques to study moisture degradation and revealed that the degradation initiated at the grain boundaries and laterally proceeded towards the grain interiors. However, information about the correlation of inner structure and degradation is rarely reported with high statistics. Therefore, it is necessary to understand the moisture degradation pathways occurring in perovskite films (in lateral and vertical directions) under an ambient atmosphere in more details. Furthermore, understanding the correlation between the moisture degradation and the

crystallographic structure features of mixed perovskite films is of great significance in improving the stability of future perovskite optoelectronic devices since the structures can be optimized by the use of advanced film fabrication methods[28,29].

Until now, spin coating has been extensively used to fabricate high-performance perovskite films. However, this method, unlike spray coating, is challenging to produce films in a way that is both, high-throughput and scalable, along with being compatible with different substrates[30]. Especially, the spray deposition process allows the small perovskite crystals that have been formed to re-dissolve and then merge into larger grains by re-crystallization, which can fabricate a dense film with micro-sized grains[31], and tunable crystalline preferred orientations. These specific structure features are important for perovskite films in terms of stability and efficiency. Therefore, spray coating, as a common industrial technique, attracts wide interest. To sufficiently unveil the underneath knowledge of perovskite materials (>10 μm beyond the grain-to-grain size[13]), thick films are necessary and can be achieved by spray coating.

In this work, we utilize nano-focused wide-angle X-ray scattering (nWAXS) to map a spray-cast $(MAPbBr_3)_{0.50}(FAPbI_3)_{0.50}$ mixed hybrid perovskite film over a large area size of 80 μm × 20 μm (length × depth) (Fig. 1a). The structure heterogeneity of the $(MAPbBr_3)_{0.50}(FAPbI_3)_{0.50}$ perovskite film is revealed with respect to the strain-related $q$ position distribution and orientation distribution (Fig. 1b). Notably, nWAXS performed in transmission geometry is able to probe local structure information at different depths within the polycrystalline film. Moreover, it is able to access the composition distribution of ambient-atmosphere moisture-induced degradation products like $PbI_2$ (Fig. 1b). Thereby, we provide an important insight into the fundamental understanding about the vertical strain distribution and orientation variations within microcrystalline domains, and emphasize the influence of structure features on the moisture degradation of perovskites.

## Results

We fabricate a mixed $(MAPbBr_3)_{0.50}(FAPbI_3)_{0.50}$ hybrid perovskite film by spray coating. To achieve a successful transmission measurement, the specimen's width needs to be ~60 μm along the incoming beam direction (Supplementary Fig. 1), and thus focused ion beam (FIB) milling is used to obtain the defined dimensions (Supplementary Fig. 2c). To characterize the local structural properties of the perovskite film at a long-range scale, we perform scanning nWAXS at the nanofocus endstation of beamline P03 of PETRA III at DESY (Hamburg, Germany)[32]. Photographs of the nWAXS setup are shown in Supplementary Fig. 2a and 2b. Spatially resolved nWAXS scans are done in 160 × 40 frames with a step-size of 500 nm (Supplementary Fig. 2c; 80 μm length × 20 μm depth), which ensures that we measure sufficient long-range features of the perovskite film in both, lateral and vertical directions. Due to the surface roughness at the air interface and the milling precision of the FIB (Supplementary Fig. 3), the nano X-ray beam directly travels through some parts of air and it is possible to visualize the surface topography of the film within an initial depth (depth <3.5 μm; details in Supplementary Figs. 4 and 5). The measured spots, which show a similar scattering signal intensity as deep inside the film, start at a depth of 3.5 μm, which is defined as the surface level (Supplementary Fig. 4). More details about the nWAXS experiments are provided in the "Methods" section.

The nWAXS study shows the characteristic features of a cubic $(MAPbBr_3)_{0.50}(FAPbI_3)_{0.50}$ structure, as well as non-negligible impurity peaks from the moisture-induced degradation products located close to the top surface. The maps of radially integrated two-dimensional (2D) nWAXS data of the spray-cast film (Supplementary Figs. 7-9) show strong Bragg peak intensities with distinct diffraction spots, which are identified by comparison with simulated XRD patterns (Supplementary Fig. 10) based on literature CIF files (see Supplementary Data 1). This is also visible in the summed 2D nWAXS patterns (scanning a large length of 80 μm) at different depths (Fig. 1c). To analyze the crystal structure, we extract radially integrated line profiles from the individual scattering patterns, which are selectively plotted in Supplementary Figs. 11-13. The dominant diffraction peaks within the film (from the surface to the bottom) are indexed, corresponding to (100), (110), (111), (200), (210), (211) and (220) reflections of the $(MAPbBr_3)_{0.50}(FAPbI_3)_{0.50}$ perovskite[21,33,34]. However, close to the top surface impurity peaks appear that are assigned to the moisture-induced degradation products, e.g., $PbI_2$[35]. In addition, we observe the weak hexagonal non-perovskite phase, known as the δ-phase of a FA⁺ rich perovskite (i.e., Supplementary Fig. 11)[36].

### Residual strain distribution in the mixed perovskite films

To probe the residual strain distribution, especially in the vertical direction, we investigate spatial $q$ maps of the main diffraction peaks from the $(MAPbBr_3)_{0.50}(FAPbI_3)_{0.50}$ perovskite film. The $q$ maps for the prominent (100), (110), (111), (200), (210), and (220) reflections (Fig. 2a) reveal the local structure heterogeneity of the mixed perovskite film on the nano-scale. As reported, strain (tensile/compressive) in a material can be determined via comparing the $q$ position of the material to a reference value (unstrained)[14]. Here, we assume the maximum $q$ position in each local distribution as the reference value, and use the relative shift of the $q$ value from the maximum $q$ (i.e., strain $= (q_{max} - q)/q_{max}$)[16]. We find that lower $q$ position values generally appear at the top region of the perovskite film (Fig. 2a; light yellow regions), while higher $q$ position values are seen at other depths (Fig. 2a; dark purple regions). To directly and statistically extract this observation, radial integrations of the summed nWAXS patterns (scanning a large length of 80 μm) at different depths are analyzed (Supplementary Fig. 14). Along with the depth increase (from the surface to the interior of the film), the subtle shifts at the peak positions (indicated with the yellow and purple dashed lines in Supplementary Fig. 14) show a similar tendency. To further quantify the

variations in $q$ positions and residual strains (Supplementary Note 2; the $q$ position at the depth of 20 μm is taken as a reference value), Bragg peaks of the radial integration profiles (Supplementary Fig. 14) are fitted with Gaussian functions. The (100), (110) and (111) $q$ positions in general increase with increasing depth (Supplementary Fig. 15), whereas the microstrains extracted from these three peaks show an opposite trend (Fig. 2b and Supplementary Fig. 16). This finding reveals that the top region of the film bears the tensile strain (Fig. 2c). To support our hypothesis (bottom region is strain-free) and finding, we further use the Williamson-Hall method[16,37] to analyze the microstrain (Supplementary Note 1 and Supplementary Fig. 17). It clearly shows that the strain in the top region is higher than that in a deep zone of the film (Supplementary Fig. 17). This finding agrees well with the literature[15,38], even though each microscale area in Fig. 2a displays its own unique local strain environment. In detail, the microstrain has a complex non-uniformity with a typical magnitude of (0.17 ± 0.15)%, as statistically estimated from the microstrain for the (100), (110) and (111) peaks in Fig. 2b, which is similar to the reported value (-0.1–0.2%)[16]. Tensile strain at the top region of the $(MAPbBr_3)_{0.50}(FAPbI_3)_{0.50}$ perovskite film is caused by the film preparation, during which the film is cooled down from a hot plate at 100 °C. The top surface of the film cools down much faster than other film regions, resulting in a lower volume shrinkage[15,39]. In addition, a lower thermal expansion coefficient of the contact layer compared to the perovskite, that leads to a tensile strain, should be considered (Supplementary Note 3)[15,38,40]. Comparing the $q$ values along the horizontal direction, no general tendency is found (Fig. 2a, Supplementary Figs. 11–13, 18). This finding might be a result of the big vertical temperature difference, which can set up strain gradients along this direction. However, the temperature variation along the lateral direction is minimized in the fabrication process. Owing to the intrinsic anisotropy properties of the mixed perovskite[21,41], the local $q$ position variations of different crystallographic planes subjected to strain (stress) are not uniform (Fig. 2a). Therefore, we reason that the $(MAPbBr_3)_{0.50}(FAPbI_3)_{0.50}$ perovskite film yields a uniaxial strain (anisotropic), with a complex local heterogeneity[16,42].

### Topological transformation of texture orientation

To understand the texture orientation and its topological transformation, we further quantitatively analyze the preferred orientation of the typical (100) perovskite peak and its related intensity (azimuthal angle χ analysis, details in "Methods" section). The preferential vertical orientation (χ = 90°; the direction normal to the substrate) is beneficial for charge carrier transport within the functional stacks assembling a potential device[43,44]. Figure 3a shows the map of crystallographic preferred orientations of the (100) peak over an $80 \times 20~\mu m^2$ (length × depth) area. The oriented angle (χ) distribution is categorized in Fig. 3e. Around 50% of (100) planes (Fig. 3e; red region) are preferentially orientated near the vertical direction. Moreover, two dominant orientations are present: One around χ of 35°, and another one χ of 64°, under the consideration of lattice distortions within the film (Fig. 3a; red and purple boxes, respectively), which is consistent with the finding reported by Pratap et al[33]. The 35° orientation corresponds to grains whose (111) planes are perpendicular to the substrate, and the 64° orientation corresponding to the (210) plane normal to the substrate (Supplementary Fig. 19). Therefore, we deduce that the major perovskite grain growth direction lies along the vertical direction[45].

Accordingly, in the spray deposited perovskite film, a longitudinal grain morphology is clearly present (Fig. 3b and Supplementary Fig. 20a). From the orientation and its corresponding intensity mapping (Fig. 3b), we can observe that numerous small grains with different orientations overwhelming appear at the bottom part, which are replaced by large grains with dominant, emergent orientations by processes akin to grain growth by annealing and ripening[31]. This

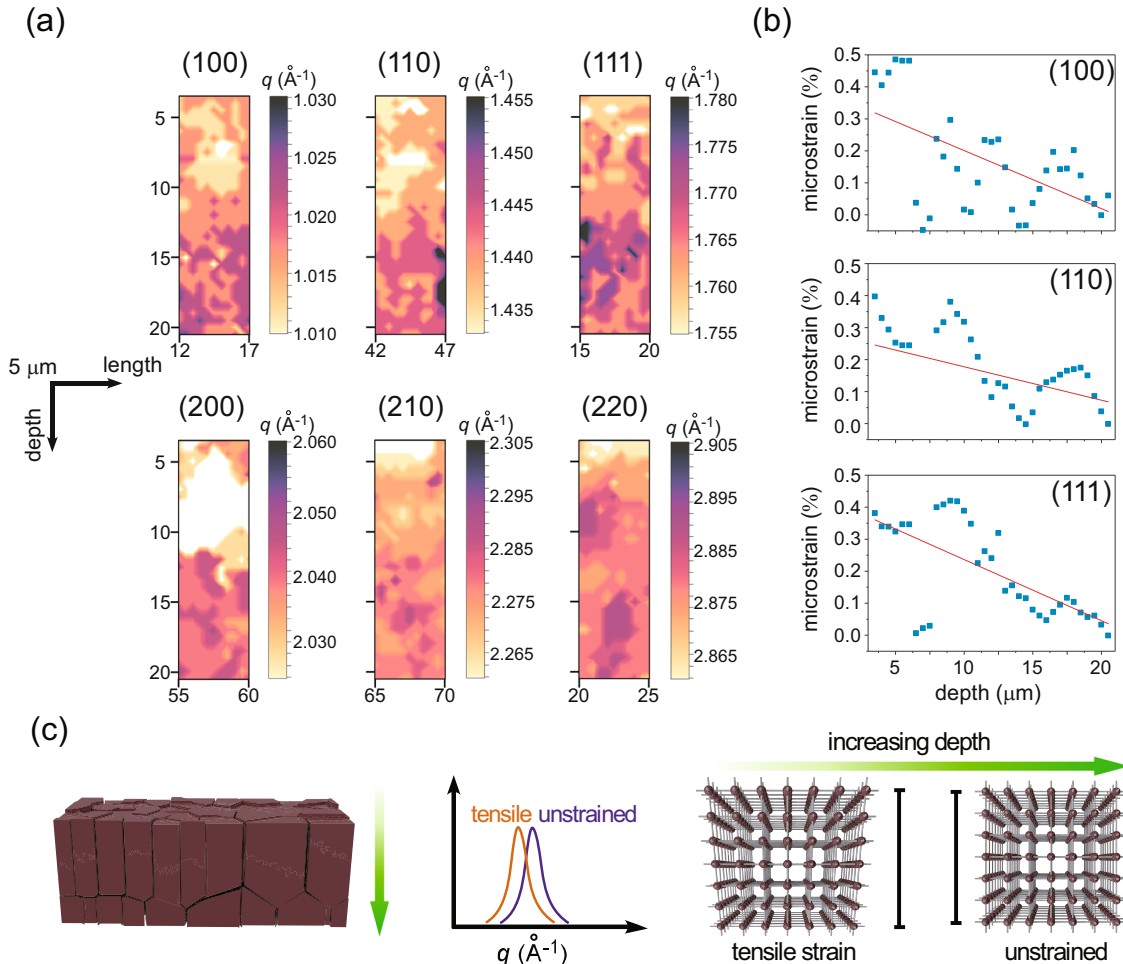

**Fig. 2 | Strain-related $q$ distribution within the perovskite film.** Selected spatial $q$ maps of the main diffraction peaks from the (MAPbBr$_3$)$_{0.50}$(FAPbI$_3$)$_{0.50}$ film: **a** (100), (110), (111), (200), (210), (220), revealing structure heterogeneity of the perovskite film. Different areas are selected, which ensures sufficient long-range features of the perovskite film. **b** Microstrain variations as a function of depth, extracted from the (100), (110), and (111) peaks. The solid line is a simple linear fit function, revealing the statistically-significant correlation of decreasing microstrain with depth. **c** Schematic illustration of the strain state in the perovskite material, demonstrating the lattice structure with the tensile strain present at the top surface of the perovskite film.

topological texture transformation agrees well with the kinetic Monte Carlo simulation reported by Gilmer et al.[46], and can be assumed as illustrated in Fig. 3f. First, near the substrate, there are initially equal distributions of a few low-energy crystallographic planes, which constitute the interface with the substrate and film surfaces. As a result, the film microstructure consists of randomly orientated or weakly textured small grains. Second, with the subsequent grain growth (also referred to as coarsening via Ostwald ripening due to solvent induced dissolution recrystallization processes), and due to anisotropies in adatom surface diffusion, the latter favored facets with more rapid vertical growth eventually dominates, e.g., (100) and (111) planes in our case, and other orientations gradually die out, resulting in the elongated-grained, textured film[47–50]. Thus, we conclude that the topological transformation of texture orientation follows the principle of evolutionary selection[49].

### Moisture-induced degradation distribution inside perovskite films

To study the structure-dependent degradation behavior of the mixed perovskite film, we analyze the spatial moisture-induced degradation distributions inside the film and especially focus on the representative decomposition product, PbI$_2$. To trace the permeation of moisture at ambient atmosphere, which will be crucial for moving perovskite materials closer towards industrial fabrications,

the PbI$_2$ (100) peak is taken as an indication. Its intensity mapping (Fig. 3c) shows that the PbI$_2$ is distributed near the top part of the perovskite film. It starts to disappear from about half of the film depth. This tendency is similar with the mapping results of the monohydrate, dihydrate and CH$_3$NH$_3$Br/CH(NH$_2$)$_2$I (Supplementary Fig. 21) being typical moisture degradation products of the perovskite. Overall, the PbI$_2$ intensity roughly ripples downward, via a layer-by-layer gradient decrease, forming the delamination of layers (Fig. 3c). Moreover, it mainly concentrates at the atmospheric interface (Fig. 3c; top left corner with the red contour). Recent studies reported that surface crystallographic defects such as vacancies or lattice distortions are sensitive to moisture and are easy to initiate perovskite decomposition[51]. In case of vacancy defects, the absorbing energies of water at vacancy sites are higher than those at the pristine surface, accelerating perovskite degradation[52,53]. Thus, we assume the following points, as sketched in Fig. 3h (type I). (1) Due to the hydrogen bond interaction with the cation or with PbI$_6$/PbBr$_6$, water can be absorbed on the surface of the perovskite film[54,55]. (2) The degradation of the cation-terminated perovskite surface could be more severe and faster, since the FA/MA ions are less resistant to moisture than Pb-I/Br bonds[24]. (3) The degradation starts at defective surface spots[51], and then undergoes expansion to surrounding areas, primarily in the lateral direction (Supplementary Fig. 20; blue box)[24].

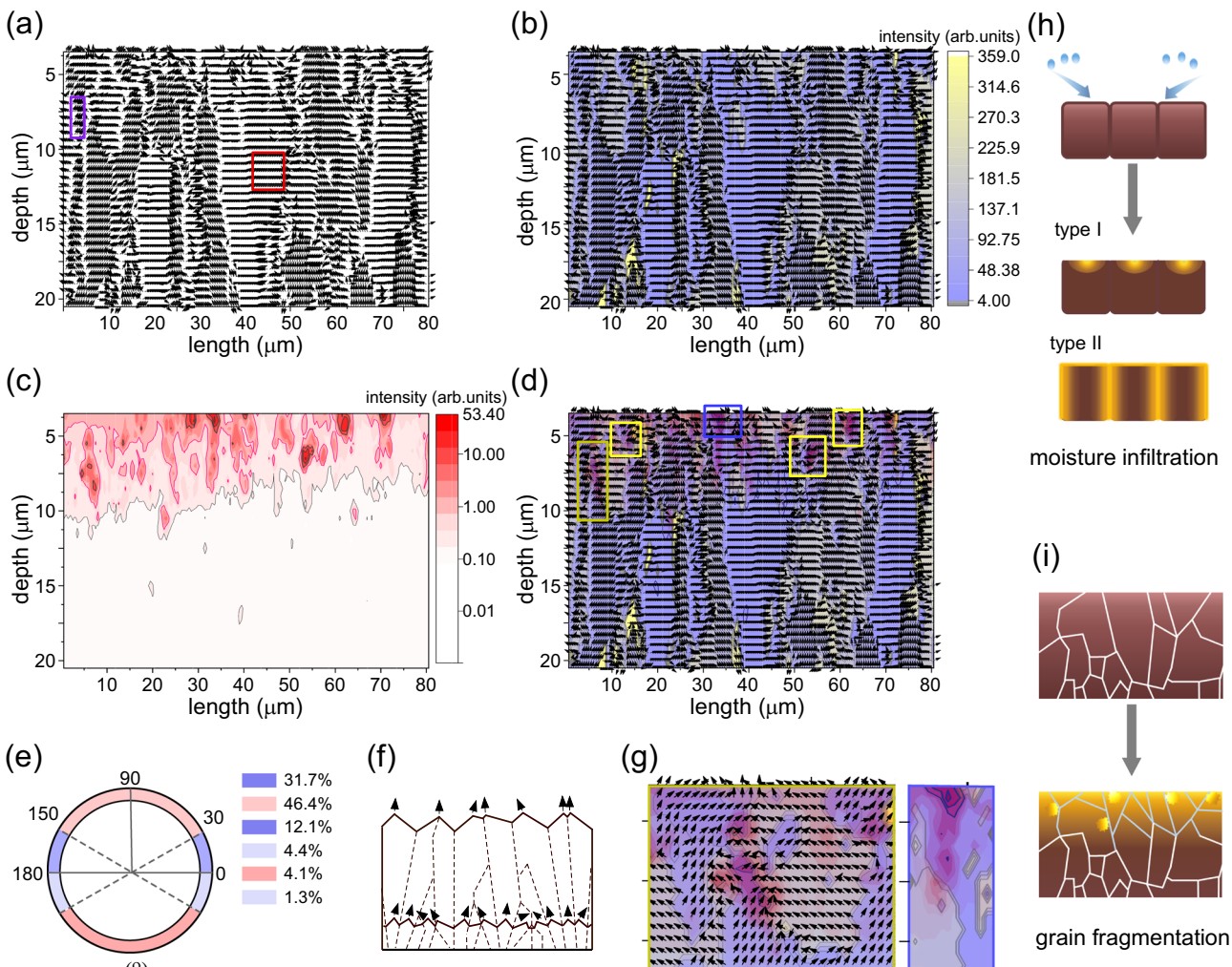

**Fig. 3 | Orientation distribution and moisture-induced degradation distribution within the perovskite film.** $80 \times 20\ \mu m^2$ (length × depth) maps of **a** preferred orientations of the (100) perovskite peak, and **b** preferred orientations and the corresponding intensity of the (100) perovskite peak. Red and purple boxes in (**a**) indicate preferred orientations of the (100) perovskite plane at around 35° and 64°. **c** $80 \times 20\ \mu m^2$ (length × depth) intensity maps of $PbI_2$ (100) peak. **d** Overlap between the orientation map of (100) perovskite peak and the intensity map of $PbI_2$ (100) peak in the same scan area. The blue and yellow boxes denote the perovskite degradation occurring at the grain surface, and at the grain boundary, respectively. **e** Schematic diagram shows the distribution of the preferred orientation angles. **f** Schematic illustration of numerous small grains with different orientations close to the bottom, replaced by large grains with dominant, emergent orientations. **g** Magnified images of the perovskite degradation region at the grain surface (blue box), and at the grain boundary (yellow box). Schematic illustration of **h** moisture degradation of the perovskite film and **i** the grain fragmentation at the degradation region.

Besides the surface, perovskite grain boundaries facilitate the quick infiltration of moisture, thus causing a rapid interior degradation of the perovskite film[24,27]. To confirm this trend, we correlate the $PbI_2$ distribution with the inner grain structure of the perovskite film. Figure 3d shows the $PbI_2$ (100) intensity mapping image overlaid with the orientation mapping of the perovskite. The intense $PbI_2$ areas with black contours (Fig. 3d; yellow boxes) occur at the sub-grain or grain boundaries, indicated from the intrinsically different local orientations[56]. Such a magnified area is clearly present in Fig. 3g (yellow box). Therefore we speculate that the grain boundary provides a route for moisture diffusion into the inner film, then the degradation initiates at the grain interface, and finally expands towards the grain interiors along the in-plane direction (Fig. 3h; type II)[24]. Note that the intrinsically local misorientations, which induce local strain, may offer a starting point for the degradation (Fig. 3d; yellow boxes)[56]. In addition, the high-intensity $PbI_2$ distribution occurs within a single grain at the surface (Fig. 3d and Supplementary Fig. 20; blue box). Its magnified image is shown in Fig. 3g (blue box). We explain such observation with the surface-defect assisted degradation mechanism[51], probably

accompanied with tensile strain[12]. To provide a perspective into the effects of strain on the perovskite degradation, we correlate the lower local $q$ positions of the (100) perovskite peak with the intensity map of the $PbI_2$ (100) peak (Supplementary Fig. 20b–d). The correlation shows that the high degradation-intensity ($PbI_2$) regions are related to the local strain centration to some extent (Supplementary Fig. 20b–d), which suggests that the strain may have an interplay relationship with moisture degradation.

To this end, the surface and grain boundaries play important roles in the perovskite stability[24–27]. However, as compared to the surface, the intense $PbI_2$ areas appear to be more present at grain boundaries (Fig. 3d and Supplementary Fig. 20). This finding indicates that grain boundaries are more easily affected by moisture, thus perhaps more critical for the stability than the surface in some perovskite films[26].

The overlap between the orientation map of the (100) perovskite peak and the intensity map of the $PbI_2$ (100) peak in the same scan area (Fig. 3d) demonstrates that the degradation region is consistent with the top part having more numerous, and smaller grains. It indicates that degradation leads to grain fragmentation, as sketched in Fig. 3i. As

mentioned above, a tensile strain exists in the top region, which can accelerate the moisture degradation, and provide a driving force for crystal fracture[40]. To reduce the free energy of the whole system, the small-sized grains break, thereby suppressing the degradation and releasing the film strain[21]. The above microstrain estimation (Fig. 2b) also shows that a rapid decrease appears at the depth of ~6.5–7.5 μm for these three planes, thus being in good agreement. Notably, the decreased ion migration caused by relaxing the strain might contribute to the suppression of degradation[11]. These critical insights into the interplay between strain and moisture degradation point out that before and after water intrusion, the residual strain in the film may be different, and a future in-depth understanding of this interplay will be essential for improving perovskite stability.

## Discussion

In summary, we investigate the local structure heterogeneity of a mixed hybrid perovskite film by scanning nWAXS, and mainly focus on the strain-related structure in terms of $q$ position distribution and orientation distribution. The relative $q$ position shifts of the $(MAPbBr_3)_{0.50}(FAPbI_3)_{0.50}$ perovskite peaks in the vertical direction reveal the presence of tensile strain at the top surface of the perovskite film. The preferred orientation map of the perovskite (100) peak identifies that the major perovskite grains grow along the vertical direction and the topological transformation of the texture orientation follows the principle of evolutionary selection. The final favored orientations depend on the crystallographic direction of the fastest vertical grain growth. Finally, we investigate the moisture-induced degradation distribution, and further correlate it to the structure features, especially the grain structure inside the perovskite film, to reveal the moisture degradation pathways. Besides the surface, at which the degradation initiates, the grain boundary allows a fast permeation of moisture into the perovskite film, causing the interior degradation of the perovskite. The tensile strain on the top surface is deemed to affect the moisture degradation. Therefore, these findings provide an in-depth understanding of the local crystal structure of the perovskite film. Based on these findings, via structure engineering, one can further optimize the moisture-resistance of perovskite films, as well as their optoelectronic properties in the future.

## Methods

### Perovskite precursor and film preparation

The 0.5 M perovskite solution was prepared by dissolving the powders methylammonium bromide ($CH_3NH_3Br$; Dyesol), formamidinium iodide ($CH(NH_2)_2I$; Sigma Aldrich), lead(II) iodide ($PbI_2$; Sigma Aldrich) and lead bromide ($PbBr_2$; Alfa Aesar) in a mixed solvent of N,N-dimethylformamide (DMF; Sigma Aldrich) and dimethylsulfoxide (DMSO; Sigma Aldrich) at a volume ratio of 4:1. The spray deposition was conducted inside a custom-made chamber in a normal atmosphere. Nitrogen with a pressure of 2 bar was applied as a carrier gas, and the substrate was placed on a heating stage with a temperature of 100 °C. To obtain the defined dimensions, focused ion beam (FIB; Zeiss NVision 40) milling was carried out on the spray-cast perovskite film at a high milling current of 1.6 nA and high acceleration voltage of 30 kV ($Ga^+$ ion source). Afterwards, the samples were stored at ambient conditions for around 30 days without special encapsulations to allow for moisture induced aging.

### Nano-focus wide-angle X-ray scattering (nWAXS)

The nWAXS experiment was executed at the nanofocus endstation of beamline P03 at PETRA III (DESY, Hamburg, Germany)[32]. Due to the absorption edge of Pb L-III (around 13.04 keV) and Br K (around 13.47 keV), the X-ray beam energy was set to 12.75 keV to minimize X-ray beam damage for the investigated perovskite film, and achieve high-transmission signals. The 2D nWAXS signals were recorded with a Pilatus 1 M detector with a pixel size $172 × 172$ μm². The sample-to-

detector distance of 244.31 mm and direct beam parameters were calibrated with the standard $LaB_6$ sample. To realize the nWAXS scanning experiment, we used a nanocube and a hexapod position to adjust the position of the investigated film in Y and Z directions (Supplementary Fig. 2b and S2c) with a well-defined step-size of 500 nm. This step-size was chosen according to the nanobeam size of 250 nm × 250 nm. The whole nWAXS scans were recorded in 160 × 40 frames, which corresponded to the scanned sample area size of 80 μm length × 20 μm depth. The two-dimensional (2D) nWAXS data were radially integrated with the DPDAK software[57]. For each spot, peak values of $q$ and the corresponding intensity were extracted. To quantify the texture orientation, the azimuthal angle $χ$ (degree) of the perovskite (100) peak were analyzed, and thus the peak of $χ$ with highest intensity was extracted. To quantify the intensity distribution of the perovskite degradation, the maximum intensities of the individual radial integration line profiles are extracted at $q$ ~0.61 Å$^{-1}$ (monohydrate), ~0.90 Å$^{-1}$ ($PbI_2$), ~1.36 Å$^{-1}$ ($CH_3NH_3Br/CH(NH_2)_2I$), and ~1.92 Å$^{-1}$ (di-hydrate).

## Data availability

All data generated or analysed during this study are included in the published article and its Supplementary Information and Source data files. The data can also be found at the following public repository: https://doi.org/10.14459/2022MP1687982.

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

## Acknowledgements

This work was supported by funding from the Deutsche Forschungsgemeinschaft (DFG, German Research Foundation) under Germany´s Excellence Strategy – EXC 2089/1 – 390776260 (e-conversion), TUM.solar in the context of the Bavarian Collaborative Research Project Solar Technologies Go Hybrid (SolTech), the Center for NanoScience (CeNS) and the International Research Training Group 2022 Alberta/Technical University of Munich International Graduate School for Environmentally Responsible Functional Hybrid Materials (ATUMS). S.P.

acknowledges support from the TUM International Graduate School of Science and Engineering (IGSSE) via the GreenTech Initiative Interface Science for Photovoltaics (ISPV) of the EuroTech Universities, the Nanosystems Initiative Munich (NIM), and the Centre for Nanoscience (CeNS). N.L. and S.L. acknowledge the China Scholarship Council (CSC). S.V. acknowledges funding from Nanosystems Initiative Munich (NIM). The authors thank Prof. Alexander Holleitner and Peter Weiser for providing access to the FIB. The authors thank Johannes Schlipf and Kiran John for help during the measurements at the beamline, and thank Wei Chen and Renjun Guo for discussion.

## Author contributions

N.L. and S.P. equally contributed to this work. N.L. wrote the manuscript and made the in-depth data analysis. S.P. and S.V. designed and executed the data management and visualization. S.P. and V.K. conducted the experiment. L.S. provided the critical insights. S.L. helped the figure design and drawing. A.D. and C.K. provided beamtime resources and support at the beamline. P.M.B. designed and supervised the project. All authors read, corrected and approved the manuscript.

## Funding

## Competing interests

The authors declare no competing interests.
