## [Peer Review File · Nature Communications]

Mapping structure heterogeneities and visualizing moisture degradation of perovskite films with nano-focus WAXSREVIEWER COMMENTS

Reviewer #1 (Remarks to the Author):

In this work, nano-focused wide-angle X-ray scattering (nWAXS) technology is developed to map the film heterogeneities over several micrometers across lateral and vertical directions. It shows that the top film region bears the tensile strain. In addition, the perovskite grains deposited by sequential spray-coating grow along the vertical direction. Moreover, the moisture-induced degradation products were revealed in the perovskite film, and the underlying mechanism for its structure-dependent degradation was elaborated. In particular, the tensile strain on the top surface has a profound influence on the moisture degradation. I believe this work did a good job to show nWAXS is a good method to visualize phase of crystallize in micrometers area. I would recommend this work to be published as long as the following concerns are well addressed.

- 1) The manuscript focuses on the influence of structure features on the moisture degradation of perovskites, but the same results were reported in other works. Please cite the related references and discuss the uniqueness of current work.
- 2) The strategies to mitigate the crystal structure heterogeneities are important. It is interesting to demonstrate the specific structure features constructed for long-term stability of PSCs with high PCE are important. Please describe the importance and unique advantages of this technology to the perovskite preparation process.
- 3) In figure 1, “close to the top surface impurity peaks appear that are assigned to the moisture-induced degradation products”, please label the specific phases of the impurity peaks according to the published literature. Moreover, how to determine if the impurity peaks are generated by water induction? Here, please add a comparison condition to exclude the effect other than water induction.
- 4) In figure 2, although the stress gradually decreases from top to bottom in the vertical direction, the stress varies greatly in both the horizontal and vertical directions. And the factors related to the stress and the regular changes in the horizontal direction should be described and the mechanisms are attractive to the audience.
- 5) The vertical changes of residual stresses are anisotropic in perovskite films, as shown in the results of (100), (110), (111) and (200), (210) (220) in Figure 2, what factors are associated with the anisotropy of this residual stress in fabrication process.
- 6) The authors claims, “a tensile strain exists in the top region, which can accelerate the moisture degradation, and provide a driving force for crystal fracture. To reduce the free energy of the whole system, the small-sized grains break, thereby suppressing the degradation and releasing the film strain”. I don't see strong evidence from N-WAXS to support this. There is no clear experimental data between the strain concentration and the decomposition starting point. It is necessary to provide some mapping data of optoelectronic properties.

7) The stress distribution with moisture degradation in the vertical direction may be different from that before the water intrusion, and is perovskite film stability affected by the surface or at grain boundaries? The author should further comment on this.

Reviewer #2 (Remarks to the Author):

This is a very good paper, whose main contribution to the literature is the quantification of strain and composition relating to degradation in perovskite thin films. To my knowledge, previous nanofocus XRD studies have been performed in a transmission geometry through the film surface, making depth-dependent information difficult to resolve. In this study, the films have been thinned down laterally allowing transmission measurements to be performed with a beam direction parallel to the surface, rastering both laterally and vertically. An understanding of degradation processes is timely and significant for perovskite films. I have not worked in the field for a while and hence I am not too familiar with the current perovskite literature, so I cannot judge well how this work dovetails with the broader literature.

The methodology for measurement and analysis is sound and appropriate. The degree of detail is sufficient for an experienced worker to reproduce. One comment I would make in this respect is that all data images presented involve integration over angle or position whereas the raw images would be in the form of a very small number of peaks, or no peaks, from the small number of illuminated grains. For Fig 1(c), in which images have been summed laterally, I wonder whether it might be helpful to provide a series of neighboring individual images in an additional figure in Supplementary Info. showing the data as measured. I would not insist on this, but it may help a diffraction worker with no experience of nanofocus work to perform similar experiments.

Reviewer #3 (Remarks to the Author):

Li et al. report a nano-focus WAXS characterization of structure heterogeneities and moisture degradation of spray-coated perovskite films. Although this nano WAXS has a limited resolution of ~500 nm, which requires a 20 μm thick spray-coated perovskite film for conducting characterizations, it is still powerful in understanding the strain distribution along thickness direction and is therefore of interest for the perovskite research community. Following are some questions and concerns; I would like to see them properly addressed in the revision process.

1. Figure 3 is a bit difficult to understand. I guess the authors use arrows in Fig. 3a to represent the crystal orientation, but these arrows are not clear. I cannot read the orientation information from Fig. 3a and cannot check the conclusion made by authors that “Red and purple boxes in (a) indicate preferred orientations of the (100) perovskite plane at around 350 and 640”. Fig. 3b, d, g also has same problem. I can understand that the authors want to present both intensity and orientation distribution in one 2D map, but both intensity and orientation are unclear for me. In Fig. 3c, the authors used a color map to present the intensity of Pbl₂, but they use a diagram with similar color to represent the orientation (Fig. 3e), which can cause potential misunderstandings. For Fig. 2a, the depth axis is clearly labelled with numbers, but the length axis has no label.

2. Regarding the depth-dependent strain shown in Fig. 2, the author find that the surface of perovskite film has a larger q and conclude that surface has tensile strain. This conclusion is made without an unstrained sample as a reference. For example, based on these data, the reader may also conclude that the bottom surface has compressive strain and top surface is unstrained. Or top surface has tensile strain and bottom surface exhibits compressive strain.

3. The strain distribution along the depth direction is fitted with a linear function, which has a large mismatch with the experimental results. For the experimental data, there are two peaks around 10 and 17 μm , respectively (Fig. 2b). Is it possible that these peaks might be caused by other reasons or just random signals originating from microscopic experiments in inhomogeneous samples. I suggest authors to present the mean values and standard deviations. Ideally, a suitable physical model should be introduced to describe the depth-dependent strain.

4. In Fig. 2a, b, the data start from the depth of $\sim 3 \mu\text{m}$, which means the signals from top surface are discarded. Although the signal from top surface is weaker than the bulk, the q of each plane can be extracted. Could the author elaborate more on this point? Figure 3 also has the same issue.

Reply to comments of Reviewer #1 (Remarks to the Author):

Comment 1: In this work, nano-focused wide-angle X-ray scattering (nWAXS) technology is developed to map the film heterogeneities over several micrometers across lateral and vertical directions. It shows that the top film region bears the tensile strain. In addition, the perovskite grains deposited by sequential spray-coating grow along the vertical direction. Moreover, the moisture-induced degradation products were revealed in the perovskite film, and the underlying mechanism for its structure-dependent degradation was elaborated. In particular, the tensile strain on the top surface has a profound influence on the moisture degradation. I believe this work did a good job to show nWAXS is a good method to visualize phase of crystallize in micrometers area. I would recommend this work to be published as long as the following concerns are well addressed.

Answer: We highly thank the reviewer for taking time to review our manuscript and giving the very positive feedback to our work.

Comment 2: The manuscript focuses on the influence of structure features on the moisture degradation of perovskites, but the same results were reported in other works. Please cite the related references and discuss the uniqueness of current work.

Answer: We thank the reviewer for the valuable comments. We have included related references about moisture degradation in the revised introduction part and added some statements on Page 3 to further highlight the uniqueness of the current work.

It reads: For instance, Wang et al.²⁴ and Yun et al.²⁷ used local real space imaging techniques to study moisture degradation and revealed that the degradation initiated at the grain boundaries and laterally proceeded towards the grain interiors. However, information about the correlation of inner structure and degradation is rarely reported with high statistics.

Comment 3: The strategies to mitigate the crystal structure heterogeneities are important. It is interesting to demonstrate the specific structure features constructed for long-term stability of PSCs with high PCE are important. Please describe the importance and unique advantages of this technology to the perovskite preparation process.

Answer: We thank the reviewer for the valuable comments. As known from the literature, spray coating involves the coalescence of droplets into a wet film (also called coarsening) followed by a drying processes. This combination allows the moistened underlying polycrystalline perovskite film with small grains to re-dissolve and then merge into larger grains by re-crystallization. As a result, a dense perovskite film with micro-sized grains can be fabricated. It means that the grain boundaries are mitigated, thereby improving the structure homogeneity. Besides, via controlling the re-dissolution and crystal grain growth, the grain orientation can also be tuned. As we wrote in the manuscript (Result part), with the subsequent grain growth, the latter favored facets with more rapid vertical growth eventually dominates, and other

orientations gradually die out, resulting in an elongated-grained, textured film. This indicates that the sub-grain crystallographic boundaries might be mitigated. These structure features are important for long-term stability and high PCE of PSCs. In the revised introduction part (Page 4), we have described the importance and unique advantages of this technology to the perovskite preparation process.

It reads:

Especially, the spray deposition process allows the small perovskite crystals that have been formed to re-dissolve and then merge into larger grains by re-crystallization, which can fabricate a dense film with micro-sized grains³¹, and tunable crystalline preferred orientations. These specific structure features are important for perovskite films in terms of stability and efficiency.

In addition, we have improved some statements in the main manuscript (Page 8).

It reads:

by large grains with dominant, emergent orientations by processes akin to grain growth by annealing and ripening³¹.

with the subsequent grain growth (also referred to as coarsening via Ostwald ripening due to solvent induced dissolution recrystallization processes),

Comment 4: In figure 1, “close to the top surface impurity peaks appear that are assigned to the moisture-induced degradation products”, please label the specific phases of the impurity peaks according to the published literature. Moreover, how to determine if the impurity peaks are generated by water induction? Here, please add a comparison condition to exclude the effect other than water induction.

Answer: We thank the reviewer for the useful comments and agree with the reviewer that we need to provide the straight details to state that the impurity peaks are generated by water induction and exclude the effect other factors than water induction. Thus, we have added an additional figure (Fig. S10) in the Supplementary Information, to directly compare the diffraction peaks in our work with those from the experimental measured data or the simulated XRD patterns (CIF files) in the published literature. Simultaneously, we have improved the details in radially integrated line profiles of 2D nWAXS data (Fig. S11-13), thereby being easily compared and consistent with Fig. S10.

It reads:

Fig. S10 Identification of the composition of the (MAPbBr₃)_{0.50}(FAPbI₃)_{0.50} film. From top to bottom, the exemplary integrated line profile from the summed scattering data at the depth of 5 μm (160 frames), experimentally measured data^{6,7} and simulated XRD patterns (CIF files; either from the published reference⁸⁻¹² or Crystallography Open Database/Cambridge Crystallographic Data Centre) are compared, to enable a clear identification of the diffraction peaks as a cubic (MAPbBr₃)_{0.50}(FAPbI₃)_{0.50} structure and the moisture-induced degradation products. The main index peaks of perovskite are indicated with black, monohydrate (CH₃NH₃PbI₃·H₂O) indicated with yellow, dihydrate ((CH₃NH₃)₄PbI₆·2H₂O) indicated with purple, PbI₂ indicated with green, CH₃NH₃Br/CH(NH₂)₂I indicated with bronze, and δ-phase indicated with blue. The colors are transparent for clarity of the presentation.

Also, we have provided the related CIF files as the Supplementary Information and added some statements on Page 20 in the main manuscript.

It reads:mappings (PDF).

Crystallographic data for $(\text{FAPbI}_3)_{0.85}(\text{MAPbBr}_3)_{0.15}$, delta FAPI, PbI_2 , $(\text{CH}_3\text{NH}_3)_4\text{PbI}_6 \cdot 2\text{H}_2\text{O}$, $\text{CH}_3\text{NH}_3\text{PbI}_3 \cdot \text{H}_2\text{O}$ and PbBr_2 (CIF).

Regarding the comment “**In figure 1, “close to the top surface impurity peaks appear that are assigned to the moisture-induced degradation products”, please label the specific phases of the impurity peaks according to the published literature.**”, we agree the reviewer that we need to label the specific phases of the impurity peaks. However, 1) we have to consider the limited space in Fig. 1; 2) We especially focus on the representative degradation product, PbI_2 in the main manuscript; 3) Fig. S11-13 in Supplementary Information clearly show the characteristic peaks from the degradation products (as seen in the improved figures above). Thus, in the revised manuscript (Page 21), we have labeled the PbI_2 (100) peak in Fig. 1 and added the related description in the figure caption.

It reads:

(c) The diffraction pattern rings, which correspond to a cubic perovskite phase, are indicated with yellow, and the diffraction ring representing the PbI₂ (100) peak is indicated with red.

Comment 5: In figure 2, although the stress gradually decreases from top to bottom in the vertical direction, the stress varies greatly in both the horizontal and vertical directions. And the factors related to the stress and the regular changes in the horizontal direction should be described and the mechanisms are attractive to the audience.

Answer: We thank the reviewer for the useful comments. In case of the horizontal direction, the q position or strain is inhomogeneous at each depth, as seen from the Fig. 2a and Fig. S11-13. For further clarification, we have provided the statistical analysis of q position variations in the horizontal direction (Fig. S18). Thus, we cannot conclude a general tendency, if we compare the q position values along the horizontal direction. This finding is similar to the results reported by Jones et al. (Fig. 1 and Fig. 3; Jones, T. W. et al. Lattice strain causes non-radiative losses in halide perovskites. *Energy Environ. Sci.* 12, 596–606 (2019)), who revealed that the strain patterns have a complex inhomogeneity.

Regarding the factors related to the stress (thermal stress involved in this work), Xue et al. (Xue, D.-J. et al. Regulating strain in perovskite thin films through charge-transport layers. *Nat. Commun.* 11, 1514 (2020)) mentioned the quantification of the stress: $\sigma_{\Delta T} = \frac{E_P}{1-\nu_P} (\alpha_S - \alpha_P) \Delta T$. E_P is the modulus of the perovskite, ν_P is Poisson's ratio in the perovskite, α_S and α_P are the thermal expansion coefficients of the substrate and the perovskite, respectively. Thus, the involved factors can be mainly divided into two categories: (1) the temperature gradient during cooling from the annealing temperature of the perovskite film to room temperature ΔT ; (2) the difference in thermal expansion coefficients (α) between the perovskite and the

contacting layers $\Delta\alpha$. Based on this equation, we explained the tensile strain at the top region (or the stress gradually decreases from top to bottom in the vertical direction) in the manuscript.

The above factors generally influence strain across the whole film. Other factors like compositional material inhomogeneity, grain boundary, etc, primarily affect the local strain.

Therefore, a uniaxial stress (strain) is present, which agrees well with other literatures (Ahn, S. M. et al. Nanomechanical approach for flexibility of organic-inorganic hybrid perovskite solar cells. *Nano Lett.* 19, 3707–3715 (2019); Jones, T. W. et al. Lattice strain causes non-radiative losses in halide perovskites. *Energy Environ. Sci.* 12, 596–606 (2019)). This might be a result of the big temperature difference along the vertical direction, especially for the thick film in this work, which can set up stress (strain) gradients along this direction. However, the temperature difference along the horizontal direction is minimized since being at the same depth.

Also considering the comment 6 below, in the revised manuscript (Page 7), we have added some description about the strain (stress) in the horizontal direction.

It reads:

.....considered (Supplementary Note 3)^{15,38,40}. Comparing the q values along the horizontal direction, no general tendency is found (Fig. 2a, Fig. S11-13 and Fig. S18). This finding might be a result of the big vertical temperature difference, which can set up strain gradients along this direction. However, the temperature variation along the lateral direction is minimized in the fabrication process. Owing to the intrinsic anisotropy properties of the mixed perovskite^{21,41}, the local q position variations of different crystallographic planes subjected to strain (stress) are not uniform (Fig. 2a). Therefore, we reason that the $(\text{MAPbBr}_3)_{0.50}(\text{FAPbI}_3)_{0.50}$ perovskite film yields a uniaxial strain (anisotropic), with a complex local heterogeneity^{16,42}.

In the Supplementary Information, we have added the q position (or strain) analysis along the horizontal direction.

It reads:

Fig. S18 Statistical analysis of q position variations in the horizontal direction. (a) Radially integrated line profiles of the vertical-summed 2D nWAXS data of the $(\text{MAPbBr}_3)_{0.50}(\text{FAPbI}_3)_{0.50}$ film (35 frames; a large depth of $\sim 20 \mu\text{m}$ is summed), (b) variations in (100), (110) and (111) peak q position as a function of length. The q profiles in (a) are fitted with a Gaussian function to extract the q positions, shown in (b). Overall, no obvious tendencies are observed along the horizontal direction, indicating that the microstrain also has a complex heterogeneity.

Finally, in the Supplementary Information, we have added an addition note.

It reads:

Supplementary Note 3. Factors related to stress (strain)

The correlation between stress (σ) and thermal expansion mismatch is quantified by³:

$$\sigma_{\Delta T} = \frac{E_P}{1-\nu_P} (\alpha_S - \alpha_P) \Delta T$$

E_P is the modulus of the perovskite, ν_P is Poisson's ratio in the perovskite, α_S and α_P are the thermal expansion coefficients of the substrate and the perovskite, respectively. The factors can be mainly divided into two categories: (1) the temperature gradient during cooling from the annealing temperature of the perovskite film to room temperature ΔT ; (2) the difference in thermal expansion coefficients (α) between the perovskite and the contacting layers $\Delta\alpha$.

For the perovskite compound, ν_P is larger than 0.3. The bulk, shear, Young's modulus ranges are within 12-30 GPa, 3-12 GPa, and 15-37 GPa, respectively. Furthermore, Young's modulus at different crystallographic planes exhibit strong anisotropic properties for all $\text{CH}_3\text{NH}_3\text{BX}_3$ ($\text{B} = \text{Sn, Pb}$; $\text{X} = \text{Br, I}$)⁴.

The above factors generally influence strain across the whole film. Other factors like compositional material inhomogeneity, phase transition, grain boundary, etc, primarily affect the local strain⁵.

Comment 6: The vertical changes of residual stresses are anisotropic in perovskite films, as shown in the results of (100), (110), (111) and (200), (210) (220) in Figure 2, what factors are associated with the anisotropy of this residual stress in fabrication process.

Answer: We thank the reviewer for the useful comments. We agree with the reviewer that the changes of the q position values are non-uniform, as shown in the results of (100), (110), (111) and (200), (210) (220) in Figure 2. This observation is consistent with the finding reported by Guo et al. (Fig. 2d; Degradation mechanisms of perovskite solar cells under vacuum and one atmosphere of nitrogen. *Nat. Energy* 6, 977–986 (2021)), who reported that the lattice shrinkage of the MAFA perovskite is non-uniform.

According to the literature (Feng, J. Mechanical properties of hybrid organic-inorganic $\text{CH}_3\text{NH}_3\text{BX}_3$ (B = Sn, Pb; X = Br, I) perovskites for solar cell absorbers. *APL Mater.* 2, 81801 (2014); Jiao, Y. et al. Strain engineering of metal halide perovskites on coupling anisotropic behaviors. *Adv. Funct. Mater.* 31, 2006243 (2021).), perovskites have very strong anisotropy. We have mentioned above, the Young's modulus at different crystallographic planes exhibit strong anisotropic properties for all $\text{CH}_3\text{NH}_3\text{BX}_3$ (B = Sn, Pb; X = Br, I). Therefore, owing to the intrinsic anisotropy properties of the mixed MAFA perovskite, the lattice parameter variations of its structures subjected to stress/strain are not uniform. These non-uniform local q position variations reflect a complex local strain heterogeneity, as reported by Jones et al. (Jones, T. W. et al. Lattice strain causes non-radiative losses in halide perovskites. *Energy Environ. Sci.* 12, 596–606 (2019)).

Factors related to the residual stress/strain, we have described in our answer to the comment 5 (see above). Thus, the anisotropy of the residual stress is significantly associated with the annealing temperature in the fabrication process and the substrate for film growth as well, in addition to the intrinsic anisotropic material properties.

Comment 7: The authors claims, “a tensile strain exists in the top region, which can accelerate the moisture degradation, and provide a driving force for crystal fracture. To reduce the free energy of the whole system, the small-sized grains break, thereby suppressing the degradation and releasing the film strain”. I don't see strong evidence from N-WAXS to support this. There is no clear experimental data between the strain concentration and the decomposition starting point. It is necessary to provide some mapping data of optoelectronic properties.

Answer: We thank the reviewer for the useful comments. We agree with the reviewer that from nano-WAXS (Fig. 3d), we can obtain that: 1) more numerous and smaller grains appear at the top region; 2) this overlaps with the degradation region. We used the reported literatures (Rolston, N. et al. Engineering stress in perovskite solar cells to improve stability. *Adv. Energy Mater.* 8, 1802139 (2018); Guo, R. et al. Degradation mechanisms of perovskite solar cells

under vacuum and one atmosphere of nitrogen. Nat. Energy 6, 977–986 (2021)) to explain these observations in our work. Our claim is strongly supported by the literature. We cited these related literatures for our explanation.

In addition, Fig. 2b shows that the microstrain decreases fast and then increase before the depth of 10 μm . Notably, there is a rapid decrease at the depth of $\sim 6.5\text{-}7.5 \mu\text{m}$, and we speculate that it may originate from the grain fragmentation.

In the revised manuscript (Page 11), we have added some statements for better explanation.

It reads:

The above microstrain estimation (Fig. 2b) also shows that a rapid decrease appears at the depth of $\sim 6.5\text{-}7.5 \mu\text{m}$ for these three planes, thus being in good agreement.

Regarding the comment “**There is no clear experimental data between the strain concentration and the decomposition starting point**”, we thank the reviewer for raising this point. First, in the revised manuscript, we have added the citation (Jariwala, S. et al. Local crystal misorientation influences non-radiative recombination in halide perovskites. Joule 3, 3048–3060 (2019)) for the related description “Note that the intrinsically local misorientations, which induce the local strain, may offer a starting point for the degradation (Fig. 3d; yellow boxes)⁵⁶”.

Besides, we have provided the local q position variations of the (100) perovskite peak reflecting the strain concentration in the Fig. S20d, according to the strain estimation (strain = $(q_{\text{max}} - q)/q_{\text{max}}$; q_{max} as a reference). We correlate the lower local q positions, derived from the perovskite (100) peak, with the intensity map of the PbI_2 (100) peak. We note that the high degradation-intensity (PbI_2) regions are relevant with the local strain concentration, at a certain degree. Importantly, the PbI_2 distribution (Fig. S20b) has been correlated with the inner grain structure of the perovskite film (Fig. S20a). Hence, these correlations indicate that the strain accelerates the degradation, and importantly, it may provide the decomposition starting point.

The systematic study of the correlation between the strain concentration and the decomposition starting point will be essential, but this study needs a series of rational designs in experiments and multi-dimensional characterizations, and computational calculation as well, to achieve a further improved in-depth understanding. Such approach clearly goes beyond the scope of the present study and could be of interest for future work.

Therefore, in the revised version, we have added Fig. S20d in the Supplementary Information.

It reads:

Fig. S20 Correlations between the moisture degradation, grain structures and the strain concentration. $80 \times 20 \mu\text{m}^2$ (length \times depth) intensity maps of (a) the preferred orientation of the (100) perovskite peak and (b) the PbI₂ (100) peak, (c) overlap between the intensity maps of the preferred orientation of the (100) perovskite peak and the PbI₂ (100) peak, and (d) local q position variations of the (100) perovskite peak reflecting the strain concentration. The depth and length are in the same scale bar. Overall, longitudinal shapes are observed in (a), indicating that the major perovskite grains grow along the vertical direction. The blue box denotes the degradation region within a single grain at the surface, and the orange boxes denote the degradation at the grain boundary. According to the strain estimation mentioned above, the lower q position reflects strain with a higher magnitude. We correlate the lower local q positions of the (100) perovskite peak (d) with the intensity map of the PbI₂ (100) peak (b). We note that the high degradation-intensity (PbI₂) regions are related to the local strain concentration at a certain degree. Hence, these correlations indicate that the strain can accelerate the degradation, and importantly, it may provide the decomposition starting point at grain boundaries or surface.

In the revised manuscript (Page 10), we have added the citation, and related description.

It reads:

Note that the intrinsically local misorientations, which induce local strain, may offer a starting point for the degradation (Fig. 3d; yellow boxes)⁵⁶.

To provide a perspective into the effects of strain on the perovskite degradation, we correlate the lower local q positions of the (100) perovskite peak with the intensity map of the PbI_2 (100) peak (Fig. S20b-d). The correlation shows that the high degradation-intensity (PbI_2) regions are related to the local strain concentration to some extent (Fig. S20b-d), which suggests that the strain may have an interplay relationship with moisture degradation.

Comment 8: The stress distribution with moisture degradation in the vertical direction may be different from that before the water intrusion, and is perovskite film stability affected by the surface or at grain boundaries? The author should further comment on this.

Answer: We thank the reviewer for raising these questions. We agree with the reviewer that before and after the water intrusion, the stress (strain) distribution in the vertical direction may be different. In the comment 5, 6 and 7, we mentioned that compositional material inhomogeneity, phase transition and grain boundary are closely relevant with the local strain. The degradation process may also involve the strain releasing.

In the revised manuscript (Page 11), we have added this point.

It reads:

These critical insights into the interplay between strain and moisture degradation point out that before and after water intrusion, the residual strain in the film may be different, and a future in-depth understanding of the interplay will be essential for improving perovskite stability.

Regarding the comment “**is perovskite film stability affected by the surface or at grain boundaries**”, we claimed that the moisture degradation initiates at the perovskite-air interface and grain boundaries. Thus, the perovskite film stability is affected by both, the surface and grain boundaries. But from Fig. 3d and Fig. S20, we can find that the intense PbI_2 areas appear to be more present at grain boundaries than at the surface. Such finding indicates that grain boundaries are more easily affected by moisture, thus perhaps more critical for the stability than the surface in some perovskite films. This is consistent with the previous literature (Sun, Q. et al. Role of Microstructure in Oxygen Induced Photodegradation of Methylammonium Lead Triiodide Perovskite Films. *Adv. Energy Mater.* 7, (2017)).

In the revised manuscript (Page 10), we have added some statements related to this topic.

It reads:

To this end, the surface and grain boundaries play an important role in the perovskite stability²⁴⁻²⁷. However, as compared to the surface, the intense PbI_2 areas appear to be more present at grain boundaries (Fig. 3d and Fig. S20). This finding indicates that grain boundaries are more easily affected by moisture, thus perhaps more critical for the stability than the surface in some perovskite films²⁶.

Reply to comments of Reviewer #2 (Remarks to the Author):

Comment 1: This is a very good paper, whose main contribution to the literature is the quantification of strain and composition relating to degradation in perovskite thin films. To my knowledge, previous nanofocus XRD studies have been performed in a transmission geometry through the film surface, making depth-dependent information difficult to resolve. In this study, the films have been thinned down laterally allowing transmission measurements to be performed with a beam direction parallel to the surface, rastering both laterally and vertically. An understanding of degradation processes is timely and significant for perovskite films. I have not worked in the field for a while and hence I am not too familiar with the current perovskite literature, so I cannot judge well how this work dovetails with the broader literature.

The methodology for measurement and analysis is sound and appropriate. The degree of detail is sufficient for an experienced worker to reproduce. One comment I would make in this respect is that all data images presented involve integration over angle or position whereas the raw images would be in the form of a very small number of peaks, or no peaks, from the small number of illuminated grains. For Fig 1(c), in which images have been summed laterally, I wonder whether it might be helpful to provide a series of neighboring individual images in an additional figure in Supplementary Info. showing the data as measured. I would not insist on this, but it may help a diffraction worker with no experience of nanofocus work to perform similar experiments.

Answer: We highly thank the reviewer for the very positive feedback. Regarding the comment about the raw images showing the data as measured, we have added a series of neighboring individual 2D nWAXS images (Fig. S6) in Supplementary Information as suggested.

It reads:

Fig. S6 Example of a series of neighboring individual 2D nWAXS images of the $(\text{MAPbBr}_3)_{0.50}(\text{FAPbI}_3)_{0.50}$ film. Selected 2D nWAXS images at the depth of 3.5 μm as a function of the sample length: 30 μm , 30.5 μm , 31 μm , 31.5 μm , 32 μm , 32.5 μm , 33 μm , 33.5 μm , and 34 μm . These images show low intensity of diffraction spots, due to the limited illuminated materials. The diffraction pattern rings, which correspond to a cubic perovskite phase, are labeled.

Reply to comments of Reviewer #3 (Remarks to the Author):

Comment 1: Li et al. report a nano-focus WAXS characterization of structure heterogeneities and moisture degradation of spray-coated perovskite films. Although this nano WAXS has a limited resolution of ~500 nm, which requires a 20 μm thick spray-coated perovskite film for conducting characterizations, it is still powerful in understanding the strain distribution along thickness direction and is therefore of interest for the perovskite research community. Following are some questions and concerns; I would like to see them properly addressed in the revision process.

Answer: We highly thank the reviewer for taking time to review our manuscript and giving the very positive feedback to our work. We have considered all comments in the revision as explained below.

Comment 2: Figure 3 is a bit difficult to understand. I guess the authors use arrows in Fig. 3a to represent the crystal orientation, but these arrows are not clear. I cannot read the orientation information from Fig. 3a and cannot check the conclusion made by authors that “Red and purple boxes in (a) indicate preferred orientations of the (100) perovskite plane at around 35° and 64°”. Fig. 3b, d, g also has same problem. I can understand that the authors want to present both intensity and orientation distribution in one 2D map, but both intensity and orientation are unclear for me. In Fig. 3c, the authors used a color map to present the intensity of PbI₂, but they use a diagram with similar color to represent the orientation (Fig. 3e), which can cause potential misunderstandings. For Fig. 2a, the depth axis is clearly labelled with numbers, but the length axis has no label.

Answer: We thank the reviewer for these useful comments to further improve the clarity of the figures. In the revised manuscript, we have made the arrows bolder and re-structured the graphical layout. Also, we have made the Fig. 3a, 3b, 3d, and 3g as big as possible, and improved the resolution.

Regarding the comment “cannot check the conclusion made by authors that “Red and purple boxes in (a) indicate preferred orientations of the (100) perovskite plane at around 35° and 64°”, we have further changed the figure size and re-structured the graphical layout of the Fig. S19 in Supplementary Information, in addition to the revision in Fig. 3 (above), to help the reviewer and readers to check more detailed information.

Regarding the comment “In Fig. 3c, the authors used a color map to present the intensity of PbI₂, but they use a diagram with similar color to represent the orientation (Fig. 3e), which can cause potential misunderstandings”, we thank the reviewer for this important comment and we agree. In the revised manuscript, we have changed the color map to avoid this problem. Accordingly, Fig. 3d, 3g and Fig. S20 (in the Supplementary Information) have been changed.

It reads:

Regarding the comment “For Fig. 2a, the depth axis is clearly labelled with numbers, but the length axis has no label”, we thank the reviewer for raising this point. The selected q area is fixed at $5 \mu\text{m} \times 20 \mu\text{m}$ (length \times depth). In the reviewed version of manuscript, we had focused on the variations in q position and residual strain along with the depth, and thus we clearly had labeled the depth axis with numbers only. To ensure that we measure sufficient long-range features of the perovskite film, different areas are selected. In the revised manuscript (Page 22), we have added the label for the length axis in Fig. 2a and some further statements for clarification.

It reads:

(a)revealing structure heterogeneity of the perovskite film. Different areas are selected, which ensures sufficient long-range features of the perovskite film.

Comment 3: Regarding the depth-dependent strain shown in Fig. 2, the author find that the surface of perovskite film has a larger q and conclude that surface has tensile strain. This conclusion is made without an unstrained sample as a reference. For example, based on these data, the reader may also conclude that the bottom surface has compressive strain and top surface is unstrained. Or top surface has tensile strain and bottom surface exhibits compressive strain.

Answer: We thank the reviewer for this important question. Indeed, more explanation is needed. To support our findings, we use the Williamson-Hall analysis (Supplementary Note 1). In Fig. S17c, it shows that the bottom region is unstrained. Consequently, from the two scenarios suggested by the reviewer, we can via the performed Williamson-Hall analysis rule out the cases conflicting with our report.

In addition, in the Supplementary Note 2, we introduced that “Here, we use $(q_{\max} - q)/q_{\max}$ to estimate the strain variation. The positions of the Bragg peaks of the perovskite film at the depth of 20 μm is taken for q_{\max} , as a reference value (without strain).”

To avoid misunderstanding, we have added some statements in the revised main manuscript on Page 6 and 7.

It reads: and residual strain (Supplementary Note 2; the q position at the depth of 20 μm is taken as a reference value),

..... To support our hypothesis (bottom region is strain-free) and finding, we further use the Williamson-Hall method^{16,37} to analyze the microstrain (Supplementary Note 1 and Fig. S17).

Comment 4: The strain distribution along the depth direction is fitted with a linear function, which has a large mismatch with the experimental results. For the experimental data, there are two peaks around 10 and 17 μm , respectively (Fig. 2b). Is it possible that these peaks might be caused by other reasons or just random signals originating from microscopic experiments in inhomogeneous samples. I suggest authors to present the mean values and standard deviations. Ideally, a suitable physical model should be introduced to describe the depth-dependent strain.

Answer: We thank the reviewer for these useful comments and suggestions. Regarding the comment “**For the experimental data, there are two peaks around 10 and 17 μm , respectively (Fig. 2b). Is it possible that these peaks might be caused by other reasons or just random signals originating from microscopic experiments in inhomogeneous samples**”, we do not think these peaks might be caused by other reasons, because we do not observe this behavior in the (100) peak. We agree with the reviewer’s latter opinion that it might be random signals originating from microscopic experiments in the mixed perovskite sample with heterogeneous structures and compositions.

Regarding the comment “**The strain distribution along the depth direction is fitted with a linear function, which has a large mismatch with the experimental results. I suggest authors to present the mean values and standard deviations. Ideally, a suitable physical model should be introduced to describe the depth-dependent strain**”, we thank the reviewer for this useful suggestion. Indeed, we used a linear function to fit a scatter plot of the data, to display a general trend of the microstrain versus the depth. We note that unveiling the nature of the depth-dependent strain requires a suitable and rigorous physical model, but this will be the subject of the future work. It deserves more attention that the current system is complicated, including the perovskite phases and degradation products. Before and after water intrusion, the residual strain in the film may be different (see comment 5-8 from reviewer 1 above). These factors might also increase the difficulty to reveal the underneath nature. Nevertheless, the statistically-significant correlation from the linear fit would not change the decreasing trend of the microstrain versus depth.

In the revised Supplementary Information (Fig. S16), we have presented the fit results showing the intercept and slope with a standard deviation and added some statements.

It reads:

Fig. S16 Statistically-significant correlation of microstrain and depth. The microstrain is estimated via the Supplementary Note 2. A simple linear fit to the scattered data reveals a statistically-significant correlation (solid red line). Notably, to unveil the nature of the depth-dependent strain, a rigorous physical model would be required and probably a simpler perovskite system would be more suitable. Also with non-linear models, the statistically-significant correlation would not change the decreasing tendency of the microstrain versus depth.

Simultaneously, in the revised main manuscript (Page 7), we have added the related description and presented the mean value and the standard deviation of the microstrain. In the figure caption of Fig. 2b (Page 22), we have also added some statements.

It reads:

.....whereas the microstrains extracted from these three peaks show an opposite trend (Fig. 2b and Fig. S16).

In detail, the microstrain has a complex non-uniformity with a typical magnitude of $(0.17 \pm 0.15)\%$, as statistically estimated from the microstrain for the (100), (110) and (111) peaks in Fig. 2b, which is similar to the reported value $(\sim 0.1-0.2\%)^{16}$.

The solid line is a simple linear fit function, revealing the statistically-significant correlation of decreasing microstrain with depth.

Comment 5: In Fig. 2a, b, the data start from the depth of $\sim 3\ \mu\text{m}$, which means the signals from top surface are discarded. Although the signal from top surface is weaker than the bulk, the q of each plane can be extracted. Could the author elaborate more on this point? Figure 3 also has the same issue.

Answer: We thank the reviewer for raising this point and we agree that the data starts from the depth of $3.5\ \mu\text{m}$ and 2D nWAXS data from top surface are not used. In the initial version of the manuscript (Page 5) and Supplementary Information (Fig. S4), we mentioned this point and explained why we treated the data like this. However, we see from the reviewer question, that our explanation was not sufficient detailed. To enable a better understanding, we have added the data at a depth of 0.5 and $3.0\ \mu\text{m}$ to the revised Supplementary Information.

It reads:

Fig. S5 Radially integrated line profiles of 2D nWAXS data of the $(\text{MAPbBr}_3)_{0.50}(\text{FAPbI}_3)_{0.50}$ film at the depth of $0.5\ \mu\text{m}$ and $3.0\ \mu\text{m}$. Note that at the depth of $0.5\ \mu\text{m}$, no intensity peaks appear in the bottom profiles, indicating that the nano X-ray beam travels through air (as shown in Fig. S4), and thus signals originate from the background. With increasing depth, the nano X-ray beam gradually illuminates the sample or partial sample. In comparison, the profiles at the depth of $3.0\ \mu\text{m}$ display numerous and clear intensity peaks. With increasing depth, each frame shows scattering signals from the sample (160 frames; $80\ \mu\text{m}$). The diffraction peaks are indexed and labeled with different colors (see the details in Fig. S10).

Simultaneously, we have improved Fig. S4 and the related description.

It reads:

Fig. S4 Mapping dimension of 80 μm length \times 20 μm depth and sample surface. (a) An illustration of the mapping dimension and sample surface. (b) A comparison of the first three frames at the depth of 3.0 μm , 3.5 μm and 4.0 μm . In (b), the scattering signal intensities of first three frames at the depth of 3.5 μm are higher than those at 3.0 μm , and comparable with those at 4.0 μm . Thus, we assume that due to the rough surface and the milling precision of the FIB, the nano X-ray beam directly travels through air or illuminates a small partial sample at the beginning of the scans (depth < 3.5 μm ; details in Fig. S5). At a depth = 3.5 μm , the measured spots (160 frames) show a similar scattering signal intensity as deeper inside the film, so that we define it as the surface level.

Accordingly, we have improved the related description in the main manuscript (Page 5).

It reads:

Due to the surface roughness at the air interface and the milling precision of the FIB (Fig. S3), the nano X-ray beam directly travels through some parts of air and it is possible to visualize the surface topography of the film within an initial depth (depth < 3.5 μm ; details in Fig. S4 and S5). The measured spots, which show a similar scattering signal intensity as deep inside the film, start at a depth of 3.5 μm , which is defined as the surface level (Fig. S4).

Thus, regarding the comment “**Although the signal from top surface is weaker than the bulk, the q of each plane can be extracted. Could the author elaborate more on this point? Figure 3 also has the same issue**”. Indeed, we agree with the reviewer that overall, the signal from top surface is weaker than from the bulk (as shown in Fig. S5). However, Fig. 3 and the description in the Method part convey that the texture orientation (χ analysis) and the intensity distribution of the perovskite degradation (q analysis) are closely related with the scattering signal intensity.

Note that no scattering peaks appear in some spots, as shown in Fig. S5, which means that we cannot extract the q of some planes. Moreover, the microstrain calculation in Fig. 2b is statistically extracted from the summed nWAXS data (160 frames, 80 μm). Thus, to obtain reliable results, especially for the mixed perovskite film showing the structure heterogeneity, the methodology for analysis in our work is appropriate.

Additional corrections:

Correction 1: Due to the change of the sequence of figures, we have made the corresponding renumbering in the entire manuscript and Supplementary Information.

Correction 2: Due to the requirements from the reviewers, we have added some addition content in the Supplementary Information. Thus, we have revised the related descriptions in the manuscript (Page 20).

It reads:

..... raw 2D WAXS images, 2D mapping images, radially integrated line profiles, diffraction indexed, scattering vector q and strain analysis,

Correction 3: Due to missing the scale bar for the intensity, we have improved the Fig. S7 in Supplementary Information.

REVIEWERS' COMMENTS

Reviewer #1 (Remarks to the Author):

The revised manuscript has been improved significantly, which provides sufficient insights for this topic. I'm happy to recommend its publication as it is.

Reviewer #2 (Remarks to the Author):

The authors have responded appropriately to referees' comments in a detailed way.

The manuscript is suitable for publication in its revised form.

Reviewer #3 (Remarks to the Author):

I am satisfied with the responses from the authors and support its publication in its current form.

Reply to comments of Reviewer #1 (Remarks to the Author):

Comment: The revised manuscript has been improved significantly, which provides sufficient insights for this topic. I'm happy to recommend its publication as it is.

Answer: We highly thank the reviewer for taking time to review our manuscript again and giving the very positive feedback to our work.

Reply to comments of Reviewer #2 (Remarks to the Author):

Comment: The authors have responded appropriately to referees' comments in a detailed way. The manuscript is suitable for publication in its revised form.

Answer: We highly thank the reviewer for taking time to review our manuscript again and giving the very positive feedback to our work.

Reply to comments of Reviewer #3 (Remarks to the Author):

Comment: I am satisfied with the responses from the authors and support its publication in its current form.

Answer: We highly thank the reviewer for taking time to review our manuscript again and giving the very positive feedback to our work.